# OPTIMAL ALGORITHM FOR MAX-MIN FAIR BANDIT

## ABSTRACT

We consider a multi-player multi-armed bandit problem (MP-MAB) where $N$ players compete for $K$ arms in $T$ rounds. The reward distribution is heterogeneous where each player has a different expected reward for the same arm. When multiple players select the same arm, they collide and obtain zero reward. In this paper, we aim to find the max-min fairness matching that maximizes the reward of the player who receives the lowest reward. This paper improves the existing regret upper bound result of $O(\log T \log \log T)$ to achieve max-min fairness. More specifically, our decentralized fair elimination algorithm (DFE) deals with heterogeneity and collision carefully and attains a regret upper bounded of $O((N^2 + K) \log T/\Delta)$, where $\Delta$ is the minimum reward gap between max-min value and sub-optimal arms. We assume $N \leq K$ to guarantee all players can select their arms without collisions. In addition, we also provide an $\Omega(\max\{N^2, K\} \log T/\Delta)$ regret lower bound for this problem. This lower bound indicates that our algorithm is optimal with respect to key parameters, which significantly improves the performance of algorithms in previous work. Numerical experiments again verify the efficiency and improvement of our algorithms.

## 1 INTRODUCTION

Multi-player multi-armed bandit (MP-MAB) problem has been widely studied in recent years (Wang et al., 2020; Yang et al., 2022; Wang et al., 2019; 2022; Yang et al., 2023). In such a problem, $N$ players simultaneously play $K$ arms. In each round, each player selects an arm and observes the reward generated from a fixed distribution. In typical MP-MAB problems, players try to maximize the summation of their cumulative expected reward throughout $T$ rounds. Equivalently, they minimize the total regret, defined as the difference of cumulative reward between their decision and the optimal strategy. MP-MAB problems can be divided into the homogeneous setting and the heterogeneous setting by whether the arm's reward varies among players. Besides, collision is also considered in many applications: when different players select the same arm in one round, they collide and all receive a 0 reward rather than the original reward drawn from the fixed distribution.

When considering the heterogeneous setting and collisions, it often raises concerns about unfairness. In order to maximize the total rewards and avoid collisions meanwhile, some players have to sacrifice for the global objective by selecting arms with lower reward (Hossain et al., 2021; Bistritz et al., 2020; Leshem, 2023). Therefore, the primitive objective which only cares about maximizing total rewards, is unfair for those players and makes them have a willingness to not follow the algorithm. Thus researchers hope to search for a better objective to avoid this unfairness. One reasonable objective is to maximize the expected reward of the player who receives the lowest reward, which is called max-min fairness (Bistritz et al., 2020; Leshem, 2023). This ensures that every player is considered equally and nobody should sacrifice. For instance, in wireless networks, when more than one users transfer information using one channel, they will interfere one another and fail, and this is modeled as collisions in our MP-MAB problem. If we only pursue the maximal Quality of Service (QoS) globally, some users will be annoyed as they can only use channels with low quality to avoid collisions of others. Therefore, fairness guarantee indeed matters. Besides, these applications also imply that we need to consider decentralized case carefully, in which players cannot gather global information directly. In the above example, a central server is able to integrate and process information in smaller networks, while in larger ones, as it costs too many resources to build a server, while in decentralized setting, the process of information propagation can be implemented by the collisions between players indirectly, which distributes resources to local players, thus, decentralized algorithm should be designed.

There are several kinds of fairness problems studied in many works. Most discussions are from the respect of arms (Joseph et al., 2016; Wang et al., 2021; Fang et al., 2022; Mansoury et al., 2021; Jeunen & Goethals, 2021), and are totally different from max-min fairness. They proposed that the arms with higher reward mean should be exposed more frequently to users and similar arms should have similar exposure frequencies. However, from the players' perspective, the concerns of unfairness still remains. Only limited papers (Bistritz et al., 2020; Leshem, 2023) studied this max-min fairness topic, but both of them have the following limitations. 1) Their regret upper bounds are larger than $O(\log T)$. The work of Bistritz et al. (2020) derives a regret lower bound of $\Omega(\log T)$. However, existing works only achieve the $O(\log T \log \log T)$ regret upper bound (Bistritz et al., 2020; Leshem, 2023). Additionally, their regret analysis also relies on a large constant, which can grow exponentially with $N$ and $1/\Delta$, where $N$ is the number of players and $\Delta$ is the minimum reward gap. 2) The lower bound $\Omega(\log T/\Delta)$ in Bistritz et al. (2020) only works for parameter $T$. Nevertheless, no lower bound for parameter $N$ or $K$ is provided, which makes the hardness of the problem still unclear. 3) They have some strong assumptions about the setting. Both assume the rewards are bounded, and the latter also assumes $K = N$. Besides, they both assume the reward means of different arms for each player to be different, which is infeasible in many scenarios. These assumptions limit the application scenarios and should be relaxed for wider applications.

In order to address the limitations above, we propose our decentralized fair elimination (DFE) algorithm to find the max-min fairness for the decentralized heterogeneous MP-MAB problem where there exists no central server to assign matching directly. We relax the assumptions we mentioned before to make our algorithm more general. Besides, it can be turned out later that this algorithm is optimal since its regret upper bound matches the problem's regret lower bound.

The algorithm makes each player run a phased elimination algorithm. In each phase, players first communicate their current reward estimation to each other. Then all players can compute the lower bound of the max-min value and then eliminates arms whose upper confidence bound (UCB) index is lower than that value. After that, players explore their remaining arms in a round-robin way and the exploration length grows exponentially with the phase. Our decentralized fair elimination algorithm has the following novelties and advantages. 1) Our algorithm adaptively explores sub-optimal arms and eliminates them efficiently, assuring that those sub-optimal arms will not be selected in the later phase. This is the key to improving the regret from $O(\log T \log \log T)$ to the optimal $O(\log T)$. 2) We design a novel exploration assignment algorithm for players at each phase. Due to the elimination procedure, each player may be left with a different set of remaining arms, leading to uneven exploration frequency between different pairs and thus making the exploration process non-trivial. Our DFE algorithm optimizes this process and guarantees that all remaining player-arm pairs will be explored once in at most $N^2 + K - N$ rounds, rather than naive $NK$ rounds. We assume $N \le K$ to guarantee all players can select arms without collisions, like most MP-MAB works. This enables the algorithm to explore each player arm pair more uniformly. We also show that we can not find another assignment procedure that achieves better result. 3) The regret analysis for this algorithm is also non-trivial. We utilize the doubling trick and the property of the regret definition for the max-min fairness problem. A tight $O((N^2 + K) \log T/\Delta)$ regret bound is obtained and we will also show that this is optimal with respect all parameters.

Besides, we also derive a tighter $\Omega(\max\{N^2, K\} \log T/\Delta)$ regret lower bound for this max-min fairness problem. Here we design an interesting special case that ensures the regret lower bounded by $\Delta$ times the selection numbers of specific $\lfloor N/2 \rfloor^2$ sub-optimal player-arm pairs. Then we show that any two of these pairs can not be selected at the same time in case of suffering a large regret. Then we construct another instance that only improves the reward of one of these pairs resulting in a better max-min value. Thus by divergence decomposition, we have that any pair must be selected $\Omega(\log T/\Delta^2)$, implying the final $\Omega(\max\{N^2, K\} \log T/\Delta)$ lower bound.

**Our Contribution.** Our contributions are outlined as follows.

1. In Section 3, we propose the Decentralized Fair Elimination algorithm (Algorithm 1), which improves the existing result of decentralized case significantly. The regret upper bound of our algorithm is $O((N^2 + K) \log T/\Delta)$. In addition, we relax some assumptions required by previous works in both settings, which allows our algorithm to be applied more widely.

2. In Section 4, we give a tight lower bound, $\Omega(\max\{N^2, K\} \log T/\Delta)$, for max-min MP-MAB problem, which is with respect to not only parameter $T$ proposed in previous work,

but also $N$, $K$ and $\Delta$. This lower bound implies that our Algorithm 1 is optimal with respect to all parameters, which closes the gap of this problem.

3. In Section 5, we implement numerical experiments as well. The numerical results show that our algorithm is effective. We also compare our algorithm with previous works, and these baselines show that our algorithm indeed improves results.

**Related Work.** There is a number of work studied MP-MAB (Wang et al., 2020; Yang et al., 2022; Wang et al., 2019; 2022; Yang et al., 2023), and the references therein. They focused on the algorithms' regret, including group regret and individual regret Wang et al. (2022); Yang et al. (2023) and communication cost. Besides, some papers discuss the homogeneous and heterogeneous settings of MP-MAB (Yang et al., 2022; Zuo et al., 2023). They studied how these two settings are different from each other. On the other hand, the existence or absence of central server are discussed in Shi et al. (2021); Mehrabian et al. (2020); Buccapatnam et al. (2015); Kolla et al. (2018). However, the fairness problem is ignored in these scenarios. As we discussed above, the potential risk of unfairness for sacrificed players should be studied, and we propose algorithms to fill this gap.

Regarding the fairness problem, there are many kinds of fairness. Totally speaking, some are for arms (Joseph et al., 2016; Wang et al., 2021; Fang et al., 2022; Mansoury et al., 2021; Jeunen & Goethals, 2021), and some are for players (Hossain et al., 2021; Bistritz et al., 2020; Leshem, 2023). Max-min fairness is a kind of fairness for players and is widely considered in wireless networks. Although Bistritz et al. (2020); Leshem (2023) proposed some algorithms to handle unfairness in max-min MP-MAB, their assumptions and results are still not tight and need to be improved. Our near-optimal algorithms with tight bounds for all four parameters dramatically improve the results.

## 2 PRELIMINARIES

We consider a multi-player multi-armed bandit problem consisting of $N$ players and $K$ arms, denoted as sets $\mathcal{N} := \{1, \ldots, N\}$ and $\mathcal{K} := \{1, \ldots, K\}$, respectively. Note that we assume $N \leq K$ to ensure that all players are able to select their arms without collisions. There are $T$ rounds. In each round $t \in [T]$, each player $i \in \mathcal{N}$ selects an arm $k_i(t) \in \mathcal{K}$, and receives a reward, denoted by $r_{i,k}(t)$. We denote all players' selections in round $t$ by the matching set $m(t) := \{(1, k_1(t)), \ldots, (N, k_N(t))\}$. When two or more players select the same arm simultaneously, a collision occurs between these players and the reward of these colliding players is $r_{i,k_i(t)}(t) = 0$; otherwise, the reward $r_{i,k_i(t)}(t)$ is generated from a 1-sub-gaussian distribution with mean $\mu_{i,k_i(t)}$. Note that the reward means are heterogeneous, i.e., $\mu_{i,k}$ can be different from $\mu_{i',k}$ for $i \neq i'$. Besides, we sometimes use player-arm pair $(i, k)$ to describe player $i$ and arm $k$ for convenience. We denote the collision indicator for player $i$ in round $t$ by $C_i(t) = \mathbb{1}\{\text{player } i \text{ suffers a collision in round } t\}$. After round $t$, each player $i$ observes reward $r_{i,k_i(t)}$ and collision indicator $C_i(t)$.

We consider the decentralized setting where there is no central platform to assign matching at each round. Each player $i$ does not know which arms are selected by others in each round. At each round $t$, each player selects the arm based on its own history observations $\{r_{s,k_i(s)}, C_i(s)\}_{s \in [t-1]}$.

The objective of players is to make a matching selection strategy to find the max-min value, i.e., to maximize the minimum reward in a matching $m$, denoted by $\gamma(m) = \min_i \mu_{i,m_i}(1 - C_i(t))$, where $m_i$ is the arm selected by player $i$ in matching $m$. We define the optimal max-min value $\gamma^*$ and optimal max-min matching $m^*$ by

$$\gamma^* = \max_m \min_i \mu_{i,m_i} \,,$$

$$m^* \in \arg\max_m \min_i \mu_{i,m_i} \,.$$

We can define the regret as the difference between the optimal max-min value $\gamma^*$ and the minimum mean value among the selected arms at each round $t$, same as Bistritz et al. (2020); Leshem (2023):

$$R(T) = \mathbb{E}\left[\sum_{t=1}^{T} \left(\gamma^* - \min_i \left\{(1 - C_i(t)) \cdot \mu_{i,k_i(t)}\right\}\right)\right].$$

## 3    DECENTRALIZED FAIR ELIMINATION ALGORITHM

In this section, we introduce the Decentralized Fair Elimination algorithm (Algorithm 1).

---

**Algorithm 1** Decentralized Fair Elimination (for player $i$)

---

1: Initialize: $\hat{\mu}_{i,k}(0) = 0, N_{i,k}(0) = 0, \text{UCB}_{i,k}(0) = \infty, \text{LCB}_{i,k}(0) = -\infty, \forall i \in [N], k \in [K]$;
    $\mathcal{P} = \{(i,k) \mid \forall i \in [N], k \in [K]\}$.
2: **for** phase $s = 1, 2, \cdots$ **do**
3:     Set $\mathcal{M}_s = \emptyset$.
4:     Compute max-min matching based on $\{\text{LCB}_{i,k}(s-1)\}_{i\in[N],k\in[K]}$, gets max-min value $\underline{\gamma}_s$;
5:     **for** $\forall j \in [N], k \in [K]$ **do**
6:         **if** $\text{UCB}_{j,k}(s-1) < \underline{\gamma}_s$ **then**
7:             Remove $(j,k)$ out of $\mathcal{P}$;
8:         **end if**
9:         **if** there doesn't exist matching $m$ which contains $(j,k)$ s.t. $\overline{\gamma}_s(m) > \underline{\gamma}_s$ **then**
10:            Remove $(j,k)$ out of $\mathcal{P}$;
11:         **end if**
12:     **end for**
13:     $\mathcal{M}_s = $ **Assign Exploration**$(\mathcal{P})$;
14:     **for** $m \in \mathcal{M}_s$ **do**
15:         Select $m_i$ for $2^s$ times;
16:         Update $\hat{\mu}_{i,k}(s), N_{i,k}(s), \text{UCB}_{i,k}(s), \text{LCB}_{i,k}(s), \forall i \in [N], k \in [K]$;
17:     **end for**
18:     Communicate by implementing algorithm 3, gets information $\text{UCB}_{i',k}(s), \text{LCB}_{i',k}(s)$ for any
        player $i'$;
19: **end for**

---

Since each player can only make decisions based on its own history observation, it is necessary for players to communicate their observations and estimations with each other to find the final equilibrium (Boursier & Perchet, 2019; Wang et al., 2019; Yang et al., 2023). The key challenge here is to control the frequency of communications as too many communications will bring more communication costs, while insufficient communications cause delayed and inaccurate decision updates. Therefore, we design a phased-based algorithm that divides $T$ rounds into epochs $s = 1, 2, \ldots$ with exponentially increasing length, while keeping the length of the communication process unchanged in each epoch. This makes sure the exploration process will dominate the game as epoch $s$ increases, and thus the regret caused by the communication process will not be the leading term.

In each epoch $s$, Algorithm 1 has three phases: elimination, exploration, and communication. In the elimination phase players compute the lower bound of max-min matching based on the received information from other players and eliminate those sub-optimal arms; after that, players select their corresponding non-eliminating arms to explore in a round-robin way. At last players share their local information in the communication phase.

For the convenience of description, we first introduce some notations related to the algorithm.

Let $N_{i,k}(s)$ denote the number of times arm $k$ is selected by player $i$ in exploration phases before the $s$ times of communication, and let $\hat{\mu}_{i,k}(s)$ be the empirical mean corresponding to $N_{i,k}(s)$. Besides, let $\text{LCB}_{i,k}(s) = \hat{\mu}_{i,k}(s) - \sqrt{\frac{6\log T}{N_{i,k}(s)}}$, and $\text{UCB}_{i,k}(s) = \hat{\mu}_{i,k}(s) + \sqrt{\frac{6\log T}{N_{i,k}(s)}}$ be terminal points of the interval which contains $\mu_{i,k}$ with high probability according to Hoeffding's inequality. $\overline{\gamma}_s(m) = \min_i \text{UCB}_{i,m_i}(s)$ is the minimum UCB index in the matching $m$.

### 3.1    ELIMINATION PHASE

In the $s$-th elimination phase, all players follow same rule to compute the lower bound of current max-min matching and to eliminate sub-optimal arms based on the latest information derived from the communication phase. Each player $i$ computes $\text{LCB}_{j,k}(s)$ and $\text{UCB}_{j,k}(s)$ for all $j \in \mathcal{N}, k \in \mathcal{K}$. Then, players compute the max-min matching and corresponding max-min value $\underline{\gamma}_s$ with respect to LCBs. We execute a threshold-based algorithm to find the max-min matching (Panagiotas et al.,

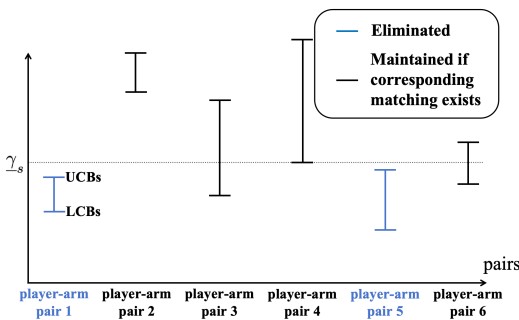

**Figure 1:** Example for elimination phase. For player-arm pairs colored blue, they will be eliminated since their UCB indexes are lower than the lower confidence bound of max-min value $\underline{\gamma}_s$.

2023). First, sort all LCB indexes $\{\text{LCB}_{i,k}(s)\}_{i\in[N],k\in[K]}$ in decreasing order, and set as threshold $\gamma'$ in order of sorting (or we can apply binary search to set threshold for reducing time complexity), testing if we can construct a perfect matching only using player-arm LCB index higher than the threshold, i.e., construct a perfect matching $m$ with $\text{LCB}_{i,m_i}(s) \geq \gamma'$. The maximum threshold we find that can form a perfect matching is the corresponding max-min value with respect to LCBs, denoted as $\underline{\gamma}_s$. For finding the max-min matching we can apply the Hungarian algorithm (Algorithm 5 in Leshem (2023)). The total time complexity is $O(\text{poly}(N, K))$, and we remark that this can be solved offline and does not affect the online learning efficiency.

After that, players eliminate the player-arm pairs who can not form a matching with minimum UCB value larger than the computed lower bound of max-min value (Lines 5 - 12). Denote $\mathcal{P}$ as the remaining non-eliminating player-arm set. If $\text{UCB}_{j,k}(s) < \underline{\gamma}_s$, then pair $(j, k)$ will be eliminated from the set $\mathcal{P}$. Additionally, if for pair $(j, k)$, we can not construct a perfect matching $m$ containing $(j, k)$ with minimum UCB index $\bar{\gamma}_s(m)$ greater than $\underline{\gamma}_s$, $(j, k)$ will also be eliminated from $\mathcal{P}$ since with high probability $(j, k)$ will not occur in the optimal max-min matching $m^*$. The details of elimination phase are described in Algorithm 1, and Figure 1 give an intuitive example for this phase.

Figure 1 gives an intuitive idea about the elimination phase. For the player-arm pairs with blue bounds in the picture, their UCBs are lower than $\underline{\gamma}_s$, so they will be eliminated. On the other hand, for the pairs with black UCBs in the picture, whether they are maintained depends on whether there's a matching, whose max-min UCB is larger than $\underline{\gamma}_s$.

## 3.2 EXPLORATION PHASE

The elimination phase is followed by the exploration phase in each epoch. Here we introduce our new assignment algorithm (Algorithm 2), which only takes at most $N^2 + K$ rounds to explore all non-eliminating pairs once under the constraint of not selecting eliminated pairs, shown to be the optimal assignment method in Section 4.

In the $s$-th exploration phase, suppose we still need to explore all player-arm pairs in $\mathcal{P}$. By the eliminating rule we have that each pair $(i, k) \in \mathcal{P}$ must have a corresponding matching $m$ such that every pair in $m$ is also in $\mathcal{P}$, i.e., $\forall (i, k) \in \mathcal{P}, \exists m, m_i = k, \forall (i', k') \in m, (i', k') \in \mathcal{P}$. Thus we can first find a matching $m'$ such that all pairs in $m'$ are in $\mathcal{P}$. Then we have a arm set $\mathcal{K}_{m'} = \{m'_1, \ldots, m'_N\}$ of all arms in $m'$. For any pair $(i, k)$ in $\mathcal{P}$, if $k \in \mathcal{K}_{m'}$, we construct a matching $m$ with $m_i = k$ and $\forall i' \in [N], (i', m_{i'}) \in \mathcal{P}$. Each $m$ is added to $\mathcal{M}$ and we have that in this process we construct at most $N^2$ matchings. After that, for those $K - N$ arms not in $\mathcal{K}_{m'}$, we give them new indexes from $N + 1$ to $K$, i.e., we extend the index of $m'_i$ to $m'_K$ such that each arm $k \in [K]$ has its unique index $u \in [K]$ with $m'_u = k$. To explore those pairs in $\mathcal{P}$ with arm not in $\mathcal{K}_{m'}$, we can construct $K - N$ matchings that explore arms in a round-robin way. Specifically, in the $r$-th matching, player $i$ will select $m'_{N+((i+r)mod(K-N))}$ if this pair is in $\mathcal{P}$, otherwise $i$ will select $m'_i$. By this design, we can ensure all pairs in $\mathcal{P}$ will be explored once in at most $N^2 + K$ rounds and all eliminated pairs are not selected. In the $s$-th exploration phase, all players select each matching for $2^s$ rounds.

---

**Algorithm 2** Assign Exploration

---

**Input:** Non-eliminating player-arm set $\mathcal{P}$.
 1: Initialize $\mathcal{M} = \emptyset$;
 2: Find a matching $m'$ satisfying $\forall (i,k) \in m', (i,k) \in \mathcal{P}$;
 3: $\mathcal{M} = \mathcal{M} \cup \{m'\}$. Denote $\mathcal{K}_{m'} = \{m'_i \mid i \in [N]\}$;
 4: **for** $\forall (i,k) \in \mathcal{P}$ **do**
 5:     **if** $k \in \mathcal{P}$ **then**
 6:        Construct a matching $m$ with $m_i = k$ and $\forall (i',k) \in m, (i',k) \in \mathcal{P}$;
 7:        $\mathcal{M} = \mathcal{M} \cup \{m\}$;
 8:     **end if**
 9: **end for**
10: Extend the index of $m'_i$ to $m'_K$ such that each arm $k \in [K]$ has its unique corresponding index $u$ satisfying $m'_u = k$;
11: Construct $K - N$ matching such that in the $r$-th matching $m^r$, player $i$ selects

$$
m_i^r = \begin{cases} m'_{N+((i+r) \bmod (K-N))}, & (i, m'_{N+((i+r) \bmod (K-N))}) \in \mathcal{P} \\ m'_i, & (i, m'_{N+((i+r) \bmod (K-N))}) \notin \mathcal{P} \end{cases}
$$

12: $\mathcal{M} = \mathcal{M} \cup \{m^r\}$ for $\forall r \in [K-N]$;
**Output:** Exploring matching set $\mathcal{M}$;

---

Figure 2 provides an example, in which blue dashed circles represent eliminated player pairs, i.e., $(i,k) \notin \mathcal{P}$. Other solid circles are available, i.e., $(i,k) \in \mathcal{P}$. When exploring pairs to the left of the vertical dashed line (arms in $\mathcal{K}_{m'}$), players can cover all of them once in at most $N^2$ rounds. When exploring pairs to the right of the vertical dashed line in a round-robin way, players select the corresponding red pair in its line if they meet a blank, i.e., an eliminated pair. For example, in Figure 2 player 2 has eliminated arm $m_{N+1}$, so when it is his turn to select $m_{N+1}$ in round-robin exploration, it will select $m_2$ instead to avoid selecting eliminated arms. Therefore, these pairs can be explored once with no more than $K - N$ rounds.

Compared with naively constructing a matching for every pair in $\mathcal{P}$ which needs at most $NK$ matchings, we improve this upper bound to $N^2 + K$. Later in the instance shown in the lower bound analysis, we will see that there exists a pair set $\mathcal{P}$ such that we can not select all pairs once below $\Omega(N^2 + K)$ rounds if we do not select pairs out of $\mathcal{P}$, which means our assignment algorithm is already near optimal.

Note that if we relax the constraint that lets players select eliminated arms, i.e, pairs out of $\mathcal{P}$, we can directly design a round-robin exploration strategy to explore all arms in $K$ rounds. However, we make such constraint that avoid selecting any eliminated pair in the exploration phase since selecting the eliminated pair leads to more regret in one round. Recall that in our regret definition, the regret in one round is the difference between $\gamma^*$ and minimum reward in selected matching. For those eliminated arms, they have been identified as sub-optimal and usually with low reward, which means we will suffer large regret if we select the eliminated arm in the exploration phase. In the regret analysis we can see that by not exploring the eliminated arm, we can improve the regret from $O(\log T/\Delta^2)$ to $O(\log T/\Delta)$, where $\Delta$ is the minimum reward gap between $\gamma^*$. This illustrates our algorithm outperforms in the scenario where there are pairs with rewards near $\gamma^*$.

### 3.3 COMMUNICATION PHASE

The communication phase has a fixed length. In $s$-phase communication, player $i$ sends $\hat{\mu}_{i,k}(s)$ for $k \in [K]$ to all other $N - 1$ players, and receives $\hat{\mu}_{j,k}(s)$ from each player $j \in [N]$ for $k \in [K]$. As $K \geq N$, there are at most $\lfloor \frac{N}{2} \rfloor$ pairs of players exchanging information meanwhile. The order of the exchange can be fixed before the game, which only needs to know the number and index of players. Here we assume that any two players can exchange their estimations of reward in one round with cost $\gamma^*$ (maximal regret in one round). Therefore, each communication phase has a fixed length of $N$.

Note that in the paper we study the problem in decentralized setting where each player can only observe its own reward and make decisions based on its historical observation. This setting is

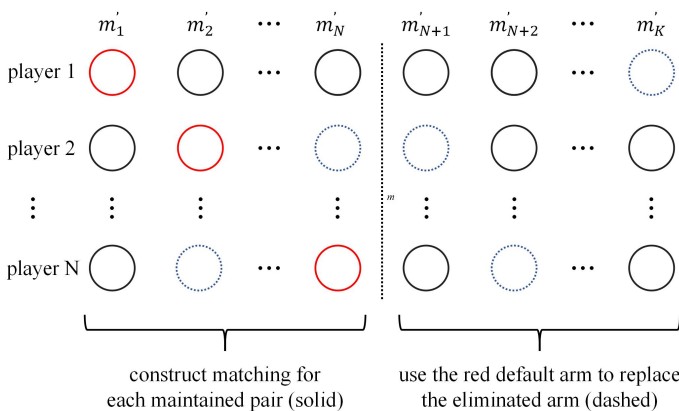

$m'_1$ $m'_2$ $\cdots$ $m'_N$ $m'_{N+1}$ $m'_{N+2}$ $\cdots$ $m'_K$

player 1

player 2

player N

construct matching for
each maintained pair (solid)

use the red default arm to replace
the eliminated arm (dashed)

**Figure 2:** Example for exploration phase. For the left $N$ arms, we construct the matching for each $(i, k)$ pairs, requiring at most $N^2$ matchings. For the right $K - N$ arms, we apply the round-robin methods and replace those eliminated arms with corresponding red solid arms to avoid collisions.

extensively studied by MP-MAB works. Furthermore, communication process among players is unavoidable since all players have to agree on a matching to avoid collisions. In this paper we control the exponential growth of communication intervals, guaranteeing that the total communication times is bounded by $O(\log T)$. Here we assume that at each round a player can transmit estimated reward information of one arm to another player. This assumption is not that strong since the amount of information transmitted in one round is bounded. Moreover, we remark that this assumption can also be removed by making players transfer the information into binary and communicate by colliding on bit "1" (Boursier & Perchet, 2019).

### 3.4 Analysis and Discussion

Now, we are ready to state our main result as the next theorem. The proof is deferred in Appendix B.

**Theorem 1.** *Algorithm 1 has expected regret bounded by*

$$\mathbb{E}[R(T)] \leq 164(N^2 + K)\log T/\Delta + 2NK + \gamma^* N \log T,$$

*where $\Delta := \min_{i,k:\mu_{i,k}<\gamma^*}\{\gamma^* - \mu_{i,k}\}$.*

*Remark 1.* We assume that each player can communicate with another player in one round at a cost of $\gamma^*$. This assumption simplifies the analysis of the communication phase. Without this assumption, we could still use collisions to transmit information bit by bit, resulting in an additional constant length of information bits. The specifics of this process can be found in Appendix F. If the minimum reward gap between max-min value $\gamma^*$ is $\Delta$, then the length of each communication phase is bounded by $N\log(1/\Delta)$. Here $\log(1/\Delta)$ is the length of transmitting a reward's information by bit and through collisions. We only need the bit length with $\log(1/\Delta)$ since it is enough to distinguish two pairs with gap larger than $\Delta$. Then the total communication cost is $N\log(1/\Delta)\log T$. We also note that the minimum communication cost studied in previous work is $\frac{3}{2}N^3\log(1/\Delta)\log T$ (Leshem, 2023). Additionally, we note that $\Delta$ in their works is the minimum reward gap among all player-arm pairs, whereas in our work $\Delta$ is only the minium reward gap between $\gamma^*$. Thus we also significantly improve the communication cost compared with previous works.

*Remark 2.* We highlight that this exploration design is optimal with order $O(N^2 + K)$. In Section 4 we will see that the given special case provides a $\Omega(N^2 + K)$ assignment lower bound when the player-arm pairs with $0$ reward are all eliminated. Thus our method indeed gives an optimal solution for such a player-arm pair covering problem without collisions. Besides, this technique can also be considered to apply for other MP-MAB problems with heterogeneous reward, which requires different exploration times for each player-arm pair to reach the optimal regret.

**Figure 3:** Base case when $N$ is even and solid circles represent max-min matching.

**Figure 4:** Increasing the value of the pair $(N, 1)$ to be optimal, the max-min matching changes.

*Remark 3.* We note that previous works Bistritz et al. (2020); Leshem (2023) also provide a phased-based algorithm to deal with the decentralized setting. However, they both apply an explore-then-commit (ETC) method at each epoch $s$. Specifically, they let each player explores each arm $\log s$ times at the beginning of the epoch $s$, and then compute the max-min matching based on history observations in exploration phase. After that each player follows this matching in the following $2^s$ rounds. Their algorithms both only obtain an $O(\log T \log\log T)$ regret bound since they have to set an increasing length of exploration at each epoch. This design is to make sure the probability of computing a wrong max-min matching is bounded by $\exp(-s)$ when $s$ is sufficiently large that $\log s > 1/\Delta$, then the regret in the exploitation phase can be bounded. This design also raises the problem of a large constant to guarantee $\log s > 1/\Delta$, which requires initial warm-up rounds is $O(\exp(1/\Delta))$, which could be very large when $\Delta$ is small enough. We handle this problem by applying the elimination method which eliminate sub-optimal player-arm pair efficiently. This assures that no forced explorations will happen in later epochs.

*Remark 4.* We claim that in the beginning of the game, there will be an index searching process if players do not know their indicies beforehand. We note that this is a standard process in decentralized MP-MAP problem Rosenski et al. (2016); Boursier & Perchet (2019) and we can run the process proposed in Leshem (2023). In this algorithm, players determine their respective indices according to a randomly choosing arm method. They show that this process will end in less than $O(N \log N)$ rounds with probability 1, therefore, the regret of this process is also less than $O(N \log N)$. Since the regret of by index searching process never dominates, for the convenience of description, we ignore it when considering the overall regret.

## 4  REGRET LOWER BOUND

In this section, we give a regret lower bound of max-min MP-MAB problems, which shows that the result of our proposed algorithm is tight with respect to parameters $K, N, \Delta$ and $T$.

Let $R(T, \nu, \pi)$ denote the expected regret of a policy $\pi$ on the instance with an arm distributions $\nu = \{\nu_{i,k} : i \in [N], k \in [K]\}$ for a horizon of length $T$. Denote $\mathcal{P}$ as the set of all probability distributions of reward bounded by $[0, 1]$.

We define a policy is *uniformly consistent* if and only if for all $\nu \in \mathcal{P}$, all $\alpha \in (0, 1)$, the regret $\limsup_{T \to \infty} \frac{R(T, \nu, \pi)}{T^\alpha} = 0$. This notion is used to eliminate tuning a policy to the current instance while admitting large regret in other instances.

We construct the special instance $\nu$ with $N < K$, the reward mean is designed as follows:

- For player $i \in [1, \lfloor \frac{N}{2} \rfloor - 1]$, the reward of arm $k \in [1, i+1]$ is $1/2 + 2\Delta$, the reward of $k \in [i+2, K]$ is 0.

- For player $i \in [\lfloor \frac{N}{2} \rfloor, N - \lfloor \frac{N}{2} \rfloor + 1]$, the reward of arm $k \in [1, i-1]$ is $1/2$, the reward of arm $k = i$ is $1/2 + \Delta$, the reward of arm $k = i + 1$ is $1/2 + 2\Delta$, and the reward of arm $k \in [i+2, K]$ is 0.

- For player $i \in [N - \lfloor \frac{N}{2} \rfloor + 2, N]$, the reward of arm $k \in [1, N - \lfloor \frac{N}{2} \rfloor + 2]$ is $1/2$, the reward of arm $k \in [N - \lfloor \frac{N}{2} \rfloor + 3, \min\{i+1, N\}]$ is $\frac{1}{2} + 2\Delta$, and the reward of $k \in [\min\{i+1, N\} + 1, K]$ is $0$.

Denote $\nu_{i,k}$ as the distribution of rewards obtained when arm $k$ is matched to player $i$ in this environment. The special case is shown in Figures 3 and 4.

In this instance, we first show that the max-min value is $\frac{1}{2} + \Delta$ with corresponding max-min matching $m^* = \{(1,1), (2,2), \ldots, (N,N)\}$, shown in Figure 3. First, we can verify the matching $m^*$ has minimum reward $\frac{1}{2} + \Delta$. Second, if there exists a matching $m'$ with minimum reward $\frac{1}{2} + 2\Delta$, then by construction the player $i \in [\lfloor \frac{N}{2} \rfloor + 1, N - \lfloor \frac{N}{2} \rfloor + 1]$ must select arm $k = i + 1$. To reach the reward $\frac{1}{2} + 2\Delta$ player $i \in [N - \lfloor \frac{N}{2} \rfloor + 2, N - 1]$ must select arm $i + 1$, and then player $i = N$ fails to select any arm with reward $\frac{1}{2} + 2\Delta$. Thus the max-min value is $\frac{1}{2} + \Delta$.

We denote the set of player-arm pairs $S_1 = \{(i,k) \mid \mu_{i,k} = 0\}$, $S_2 = \{(i,k) \mid i \in [N - \lfloor \frac{N}{2} \rfloor + 1, N], k \in [1, \lfloor \frac{N}{2} \rfloor]\}$ (gray shaded area in Figure 3). When selecting a player-arm pair in $S_1$ will suffer a large constant regret $\frac{1}{2} + \Delta$. We claim that when not selecting any pair in $S_1$, any two pairs in $S_2$ will not be in the same matching:

**Claim 1.** *Given a matching $m$, if $\forall (i,k) \in S_1, (i,k) \notin m$, then $\forall (i,k), (i',k') \in S_2$ with $(i,k) \neq (i',k')$, we have $(i,k) \notin m$ or $(i',k') \notin m$.*

*Proof.* We show this claim by contradiction. Consider a matching $m$ that $\forall (i,k) \in S_1, (i,k) \notin m$. Suppose $\exists (i,k), (i',k') \in m$ with $i < i'$, $(i,k) \in S_2$ and $(i',k') \in S_2$. From the construction of $S_1$ and instance $\nu$, we have that for player $i'' \leq \lfloor \frac{N}{2} \rfloor$, it can only select arm with index less than $\lfloor \frac{N}{2} \rfloor + 1$. Since $i' > i > \lfloor \frac{N}{2} \rfloor$, we have that the number of remaining arms available for top $\lfloor \frac{N}{2} \rfloor$ players is $\lfloor \frac{N}{2} \rfloor - 1$. This contradicts that $m$ is a matching. $\square$

This claim shows that in order to explore all pairs in $S_2$ without selecting low reward pairs in $S_1$, we have to construct matching for each pair in $S_2$ separately. In other words, at least $\lfloor \frac{N}{2} \rfloor^2$ matchings are needed in exploration. Then we can have the following lemma, which is proved in Appendix D:

**Lemma 1.** *For the above instance $\nu$, and any uniformly consistent policy $\pi$, for $\Delta < 1/N$, the following holds:*

$$R(T, \nu, \pi) \geq \Delta \sum_{i=N-\lfloor \frac{N}{2} \rfloor + 1}^{N} \sum_{k=1}^{\lfloor \frac{N}{2} \rfloor} \mathbb{E}[N_{i,k}(T)],$$

*where $N_{i,k}(T)$ is the number of times $(i,k)$ is selected up to time $t$.*

For any pair $(i,k)$ in $S_2$, we can show that when $\mu_{i,k}$ is changed to $\mu'_{i,k}$ higher than $\frac{1}{2} + \Delta$, then the max-min value will be changed to $\mu'_{i,k}$. Figure 4 shows the corresponding max-min matching in this instance. This motivates us to design another instance $\nu'$ which only changes the distribution of $(i,k)$. Then applying the standard technique to lower bound $N_{i,k}(T)$ we can have the following theorem.

**Theorem 2.** *For max-min MPMAB problem with $N$ players, $K$ arms, and time horizon $T$, for a instance $\nu$ with minimum gap $\Delta < 1/N$, for any uniformly consistent policy $\pi$ satisfies*

$$\liminf_{T \to \infty} \frac{R(T, \nu, \pi)}{\log T} \geq \sum_{i=N-\lfloor \frac{N}{2} \rfloor + 1}^{N} \sum_{k=1}^{\lfloor \frac{N}{2} \rfloor} \frac{\Delta}{D_{\inf}(\nu_{i,k}, \frac{1}{2} + \Delta, \mathcal{P})}.$$

*In other words, for max-min MP-MAB problem with $N$ players, $K$ arms, and time horizon $T$, there exists an instance that for any uniformly consistent policy, for minimum gap $\Delta < 1/N$, it takes at least $\Omega(\max\{N^2, K\} \log T/\Delta)$ regret, where $\Delta = \min_{(i,k): \gamma^* > \mu_{i,k}} \{\gamma^* - \mu_{i,k}\}$.*

The detail of the lower bound proof is deferred to Appendix C.

*Remark.* We note that Bistritz et al. (2020) first derived an $\Omega(\log T/\Delta)$ lower bound of max-min problem. However, that is a loose result that does not underscore the roles of $K$ and $N$. By contrast,

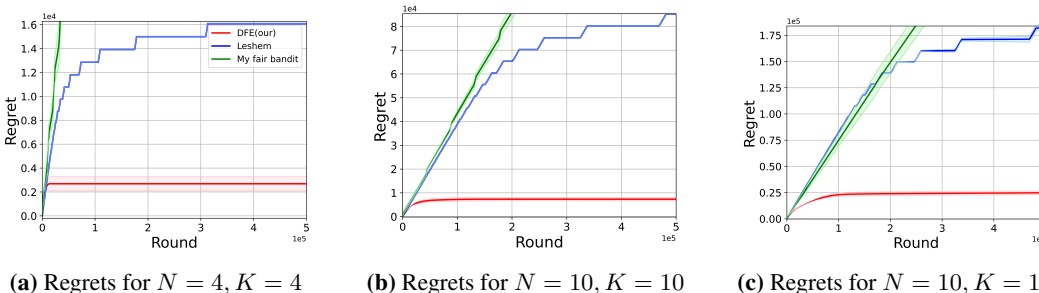

**(a)** Regrets for $N = 4, K = 4$     **(b)** Regrets for $N = 10, K = 10$     **(c)** Regrets for $N = 10, K = 15$

**Figure 5:** Experimental comparisons of our DFE algorithm with Leshem and My Fair Bandit algorithms under $(N, K) = (4, 4), (10, 10), (10, 15)$ settings respectively.

we give a tighter lower bound in Theorem 2, which works for all related parameters. In Section 3, we see that the regret of the algorithm proposed in this paper matches such lower bound. It means that we not only improve the theoretical result, but also close the gap of this problem.

## 5 EXPERIMENTS

In this section, we provide some numerical simulations to validate that our DFE algorithm performs well in max-min MP-MAB problems, besides, the comparison with previous works verifies significant improvements of our algorithm.

Here, we take $T = 500,000$, and change the values of $N$ and $K$ to implement multiple experiments: $(N, K) = (4, 4), (10, 10), (10, 15)$. The reward of each player-arm pair $(i, k)$ follows a Gaussian distribution $\mathcal{N}(\mu_{i,k}, \sigma^2)$ with $\sigma = 1$. The reward means form a reward matrix $U$, whose element in the $i$-th row and the $k$-th column is $\mu_{i,k}$. The reward matrix of $(4, 4)$ and $(10, 10)$ are the same with which in Bistritz et al. (2020), shown in Appendix E. And for $(10, 15)$, we generate the mean value uniformly from $[0, 1]$. We conducted experiments with three algorithms for comparison: DFE (Algorithm 1), Leshem ((Leshem, 2023)), and My Fair Bandit ((Bistritz et al., 2020)). As for the choice of hyperparameters, we note that our algorithm is parameter-free, and the hyperparameters of the benchmarks are the same as they stated in their paper. Each experiment was repeated 20 times. All plots are averaged over 20 trials with confidence intervals of 95%.

Figure 5 shows that our DFE algorithm dramatically decreases the regret compared to the other two algorithms. Specifically, in all three experiments, The regret of the DFE algorithm decreases more than 90% of My Fair Bandit algorithm and more than 80% of the Leshem algorithm. In addition, the DFE algorithm converges quickly, on the other hand, the other two algorithms may experience a long time before convergence, as Figure 5b shows. Besides, the other two algorithms both suffer regrets continuously since they apply the phased ETC method which requires lasting explorations even though each player has enough exploration, while our algorithm would stop selecting sub-optimal pairs when they are eliminated. Additionally, we note that our novel exploration assignment method which explores not eliminated pairs nearly uniformly also helps the algorithm attain less regret, as the other two algorithms simply explore all arms even if they have been identified to be sub-optimal. These numerical results verify the advantage of our algorithm.

## 6 CONCLUSION

This paper discusses the max-min MP-MAB problem and proposes a new algorithm with a tight regret upper bound, which effectively solve the problem of unfairness. One limitation of our work is that we don't sufficiently consider the willingness of players, in other words, if some malicious players disagree with the strategy, they may not follow the algorithm, then the whole process may fail. This implies that future work can focus on the robustness and incentive compatibility of the max-min MP-MAB problem. Another promising future direction is that our insights into the exploration stage in our DFE algorithm can be generalized into other kinds of MP-MAB problems, which serves as an effective technique dealing with collisions.

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

APPENDIX

## A    NOTATION

In this section, we summarize all used notations with their meanings in the following table.

| | |
|---|---|
| $N$ | Number of players |
| $K$ | Number of arms |
| $T$ | Number of rounds |
| $k_i(t)$ | Arm player $i$ selects at round $t$ |
| $r_{i,k}(t)$ | Reward of player $i$ selecting arm $k$ at round $t$ |
| $\mu_{i,k}$ | Mean of reward player $i$ selecting arm $j$ |
| $C_i(t)$ | Collision indicator of player $i$ at time $t$ |
| $m$ | Matching |
| $\mathcal{M}$ | Set of matching |
| $\gamma(m)$ | Minimum mean reward in matching $m$ |
| $\gamma^*$ | Max-min mean reward |
| $m^*$ | Matching with max-min mean reward |
| $\hat{\mu}_{i,k}(t)$ | Empirical mean reward of $i$ selecting $k$ at $t$ |
| $N_{i,k}(t)$ | Number of times player $i$ selects arm $k$ up $t$ |
| $\text{UCB}_{i,k}(t)$ | UCB index of player $i$ for arm $k$ at round $t$ |
| $\text{LCB}_{i,k}(t)$ | LCB index of player $i$ for arm $k$ at round $t$ |
| $\underline{\gamma}_s(m)$ | Minimum LCB value at phase $s$ in matching $m$ |
| $\overline{\gamma}_s(m)$ | Minimum UCB value at phase $s$ in matching $m$ |
| $\underline{\gamma}_s$ | Max-min LCB index at phase $s$ |

**Table 1:** Notation table that includes all used notations with their meanings in the paper.

## B    PROOF FOR SECTION 3

In this section, we first state the supplementary lemma (proved in Appendix D) for Theorem 1, and then prove this theorem.

**Lemma 2.** *Let $\mathcal{F} = \{\exists i \in [N], \exists k \in [K], |\hat{\mu}_{i,k}(t) - \mu_{i,k}| > \sqrt{\frac{6\log(T)}{N_{i,k}(t)}}\}$ be the bad event that some player-arm rewards are not estimated well at time $t$. We have:*

$$\mathbb{P}\left(\mathcal{F}\right) \leq 2NK/T \,.$$

### B.1    PROOF FOR THEOREM 1

*Proof.* First, from algorithm design we have that the regret is caused by the communication phase and exploration phase respectively. Then we have

$$\mathbb{E}\left[R(T)\right] = \mathbb{E}\left[R_{comm}(T)\right] + \mathbb{E}\left[R_{expl}(T)\right] \,.$$

Recall the definition of $\mathcal{F}$, under event $\urcorner\mathcal{F}$ we can imply that $\text{UCB}_{i,k} \geq \mu_{i,k} \geq \text{LCB}_{i,k}$. And the regret caused by exploration can be bounded by

$$\mathbb{E}\left[R_{expl}(T)\right] \leq \mathbb{E}\left[R_{expl}(T) \mid \urcorner\mathcal{F}\right] + T\mathbb{P}\left(\mathcal{F}\right) \,.$$

For each epoch $s$, we define the set of player-arm pair $\mathcal{D}_s$ as the eliminated pair at epoch $s$.

For arm $(i,k) \in \mathcal{D}_s$, we have that it is not eliminated at epoch $s-1$, where each non-eliminated arm including $(i,k)$ has been selected at least $2^s$ number of times. Then conditioned on good event $\urcorner\mathcal{F}$, we have that

$$|\hat{\mu}_{i,k}(s) - \mu_{i,k}| \leq \sqrt{\frac{6\log T}{2^s}} \,.$$

Moreover, since the optimal player-arm pair (i', k') with max-min reward $\gamma^*$ is not eliminated, it is also selected at least $2^s$ number of times, we have that

$$|\hat{\mu}_{i',k'}(s) - \mu_{i',k'}| \leq \sqrt{\frac{6 \log T}{2^s}}.$$

Since sub-optimal pair $(i,k)$ is not eliminated, we have that $2\sqrt{\frac{6 \log T}{2^s}} \geq \mu_{i',k'} - \mu_{i,k} := \Delta_{i,k}$. Otherwise it must hold that $\text{UCB}_{i,k}(s) < \text{LCB}_{i',k'}(s)$ and (i,k) will be eliminated after $s-1$ epoch.

Thus conditioned on $\neg\mathcal{F}$, we have that

$$2^s \leq 24 \log T / \Delta_{i,k}^2.$$

This implies that $\Delta_{i,k} \leq \sqrt{\frac{24 \log T}{2^s}}$, where $\Delta_{i,k} := \gamma^* - \mu_{i,k}$. Denote $n_{i,k}(T) = \sum_{t=1}^{T} \mathbb{1}\left\{(i,k) \in \arg\min_{(j,\ell) \in m_t} \mu_{j,\ell}\right\}$ as the number of times sub-optimal pair $(i,k)$ counts for the regret. Following the algorithm design, we know that the total rounds up to epoch $s$ is at most $(N^2 + K)2^{s+1}$, thus

$$\sum_{(i,k) \in \mathcal{D}_s} n_{i,k}(T) \leq (N^2 + K)2^{s+1}.$$

Denote $s_{\max}$ as the last epoch that eliminates sub-optimal pairs. We have that

$$s_{\max} \leq \log\left(24 \log T / \Delta^2\right).$$

Thus regret can be decomposed as

$$\mathbb{E}\left[R_{expl}(T) \mid \neg\mathcal{F}\right] = \sum_{(i,k)} n_{i,k}(T)\Delta_{i,k}$$

$$= \sum_{s=1}^{s_{\max}} \sum_{(i,k) \in \mathcal{D}_s} n_{i,k}(T)\Delta_{i,k}$$

$$\leq \sum_{s=1}^{s_{\max}} (N^2 + K)2^{s+1}\sqrt{\frac{24 \log T}{2^s}}$$

$$= \sum_{s=1}^{s_{\max}} 4(N^2 + K)\sqrt{6 \log T} 2^{\frac{s}{2}}$$

$$\leq 4(N^2 + K)\sqrt{6 \log T} \frac{\sqrt{2}(1 - \sqrt{2}^{s_{\max}})}{1 - \sqrt{2}}$$

$$\leq 4(N^2 + K)\sqrt{6 \log T} \frac{\sqrt{2}(1 - \sqrt{24 \log T / \Delta^2})}{1 - \sqrt{2}}$$

$$\leq 164(N^2 + K) \log T / \Delta.$$

The regret for the communication phase is bounded by the communication length times the number of phase. Since the length of phase grows exponentially we have that the total number of phase is less than $\log T$. Then the communication regret is

$$\mathbb{E}\left[R_{comm}(T)\right] \leq \gamma^* \frac{N(N-1)}{2\lfloor \frac{N}{2} \rfloor} \log T \leq \gamma^* N \log T.$$

Together with above results, we get the final regret bound. $\qquad\square$

## C    PROOF FOR SECTION 4

We construct the special instance $\nu$ with $N \leq K$, the reward mean is designed as follows: For player $i \in [1, N-2]$, the reward mean of arms $k \in [i+2, N]$ is 0. For arm $k > N$, the reward mean

is 0 for every player $i \in [N]$. For player $i \in [1, \lfloor \frac{N}{2} \rfloor]$, the reward mean of arms $k \in [1, i+1]$ is $\frac{1}{2} + 2\Delta$. For player $i \in [\lfloor \frac{N}{2} \rfloor, N - \lfloor \frac{N}{2} \rfloor + 1]$, the reward mean of arm $k = i$ is $\frac{1}{2} + \Delta$, the reward mean of arm $k = i+1$ is $\frac{1}{2} + 2\Delta$. For player $i \in [N - \lfloor \frac{N}{2} \rfloor + 2, N]$, the reward mean of arm $k \in [N - \lfloor \frac{N}{2} \rfloor + 3, \min\{i+1, N\}]$ is $\frac{1}{2} + 2\Delta$. Other player-arm pairs have reward means $\frac{1}{2}$. Denote $\nu_{i,k}$ as the distribution of rewards obtained when arm $k$ is matched to player $i$ in this environment. The special case is shown in Figures 3 and 4.

We denote the set of player-arm pairs $S_1 = \{(i,k) \mid \mu_{i,k} = 0\}$, $S_2 = \{(i,k) \mid i \in [N - \lfloor \frac{N}{2} \rfloor + 1, N], k \in [1, \lfloor \frac{N}{2} \rfloor]\}$.

We first show that the max-min value is $\frac{1}{2} + \Delta$. First, we can find a matching $m^* = \{(1,1), (2,2), \ldots, (N,N)\}$ with minimum reward $\frac{1}{2} + \Delta$. Second, if there exists a matching $m'$ with minimum reward $\frac{1}{2} + 2\Delta$, then by construction the player $i \in [\lfloor \frac{N}{2} \rfloor + 1, N - \lfloor \frac{N}{2} \rfloor + 1]$ must select arm $k = i + 1$. To reach the reward $\frac{1}{2} + 2\Delta$ player $i \in [N - \lfloor \frac{N}{2} \rfloor + 2, N - 1]$ must select arm $i + 1$, and then player $i = N$ fails to select any arm with reward $\frac{1}{2} + 2\Delta$. Thus the max-min value is $\frac{1}{2} + \Delta$.

## C.1 Proof for Theorem 2

*Proof.* For term $N_{i,k}(T)$, we apply the basic technique in lower bound proof. We consider the above instance $\nu$, universally consistent policy $\pi$, player-arm $(i,k) \in S_2$. Let us consider another instance $\nu'$ (which is specific to player $i$ and arm $k$) where $\nu'_{i',k'} = \nu_{i',k'}$ for all $(i', k') \neq (i, k)$, $\nu'_{i,k}$ such that $D(\nu_{i,k}, \nu'_{i,k}) \leq D_{\inf}(\nu_{i,k}, \frac{1}{2} + \Delta, \mathcal{P}) + \epsilon$ and $\mu'_{i,k} > \frac{1}{2} + \Delta$. Here $D_{\inf}(\nu, \mu, \mathcal{P}) = \inf_{\nu' \in \mathcal{P}}\{D(\nu, \nu') : \mu'_{i,k} > x\}$. Then the max-min value is $\mu'_{i,k}$ and the corresponding max-min player-arm is $(i, k)$.

For any event $A$ (and its complement $A^c$), applying Pinsker's inequality we have

$$D(\mathbb{P}_{\nu,\pi}, \mathbb{P}_{\nu',\pi}) \geq \log\left(\frac{1}{2(\mathbb{P}_{\nu,\pi}(A) + \mathbb{P}_{\nu',\pi}(A^c))}\right).$$

Now consider $A = \{N_{i,k}(T) \geq T/2\}$. Thus we have the regret:

1. In instance $\nu$ as $R(T, \nu, \pi) \geq \Delta \frac{T}{2} \mathbb{P}_{\nu,\pi}(A)$.

2. In instance $\nu'$ as $R(T, \nu', \pi) \geq (\mu'_{i,k} - (\frac{1}{2} + \Delta)) \frac{T}{2} \mathbb{P}_{\nu',\pi}(A^c)$.

As the only change in reward distribution happens in $(i, k)$ pair, from the divergence decomposition lemma (Lemma 18 in Sankararaman et al. (2021)), we have that

$$D(\mathbb{P}_{\nu,\pi}, \mathbb{P}_{\nu',\pi}) = D(\nu_{i,k}, \mu'_{i,k}) \mathbb{E}_{\nu,\pi}[N_{i,k}(T)] \leq \left(\epsilon + D_{\inf}(\nu_{i,k}, \frac{1}{2} + \Delta, \mathcal{P}) \mathbb{E}_{\nu,\pi}\right)[N_{i,k}(T)].$$

Then we have

$$\left(\epsilon + D_{\inf}(\nu_{i,k}, \frac{1}{2} + \Delta, \mathcal{P}) \mathbb{E}_{\nu,\pi}\right)[N_{i,k}(T)] \geq \log\left(\frac{1}{2(\mathbb{P}_{\nu,\pi}(A) + \mathbb{P}_{\nu',\pi}(A^c))}\right)$$

$$\geq \log\left(\frac{T \min(\mu'_{i,k} - (\frac{1}{2} + \Delta), \Delta)}{4(R(T, \nu, \pi) + R(T, \nu', \pi))}\right).$$

The final inequality holds as the policy $\pi$ is assumed to be universally consistent. Thus we have

$$\lim_{\epsilon \to 0} \lim_{T \to \infty} \in \frac{\mathbb{E}_{\nu,\pi}[N_{i,k}(T)]}{\log T} \geq \lim_{\epsilon \to 0} \frac{1}{\epsilon + D_{\inf}(\nu_{i,k}, \frac{1}{2} + \Delta, \mathcal{P})} = \frac{1}{D_{\inf}(\nu_{i,k}, \frac{1}{2} + \Delta, \mathcal{P})}.$$

For $\mathcal{P}$ be the class of Bernoulli rewards, we have $D_{\inf}(\nu_{i,k}, \frac{1}{2} + \Delta, \mathcal{P}) \leq \Delta^2/2$. Then we have $\mathbb{E}[N_{i,k}(T)] \geq 2\log T/\Delta^2$. And thus the total regret is lower bounded by

$$R(T) \geq \left(\lfloor \frac{N}{2} \rfloor\right)^2 \frac{2\log T}{\Delta}.$$

As for the $\Omega(K \log T / \Delta)$ lower bound, we can simply let $N = 1$ and this problem is reduced to the classic single-player multi-armed bandit problem, which has $\Omega(K \log T / \Delta)$ lower bound.

$\square$

# D  PROOF FOR TECHNICAL LEMMAS

**Lemma 3.** *(Corollary 5.5 in Lattimore & Szepesvári (2020)) Assume that $X_1, X_2, \ldots, X_n$ are independent, $\sigma$-subgaussian random variables centered around $\mu$. Then for any $\varepsilon > 0$,*

$$\mathbb{P}\left(\frac{1}{n}\sum_{i=1}^{n} X_i \geq \mu + \varepsilon\right) \leq \exp\left(-\frac{n\varepsilon^2}{2\sigma^2}\right), \quad \mathbb{P}\left(\frac{1}{n}\sum_{i=1}^{n} X_i \leq \mu - \varepsilon\right) \leq \exp\left(-\frac{n\varepsilon^2}{2\sigma^2}\right).$$

## D.1  PROOF FOR LEMMA 1

We also divide $T$ rounds into two parts: if any pair in $S_1$ occurs in $m(t)$, or there is an empty matching pair, then $t \in \mathcal{T}_1$. If $m(t)$ contains no $S_1$ pairs but contains any $S_2$ pairs, then $t \in \mathcal{T}_2$. It is easy to verify that $\mathcal{T}_1$ and $\mathcal{T}_2$ do not intersect. Denote $T_1 = \mathbb{E}\left[|\mathcal{T}_1|\right]$ and $T_2 = \mathbb{E}\left[|\mathcal{T}_2|\right]$.

$$
\begin{aligned}
R(T, \nu, \pi) &= \mathbb{E}\left[\sum_{t=1}^{T}\left(1 - \min_i\left\{(1 - C_i(t)) \cdot \mu_{i,k_i(t)}\right\}\right)\right] \\
&\geq (\frac{1}{2} + \Delta)\mathbb{E}\left[\sum_{t=1}^{T}\mathbb{1}\{t \in \mathcal{T}_1\}\right] + \Delta\mathbb{E}\left[\sum_{t=1}^{T}\mathbb{1}\{t \in \mathcal{T}_2\}\right] \\
&= (\frac{1}{2} + \Delta) \cdot T_1 + \Delta \cdot T_2.
\end{aligned}
\tag{1}
$$

Before analyzing term $T_2$, we first claim the following property of matching when no pair in $S_1$ is selected.

For term $T_2$, recall that it means the number of times $m(t)$ contains no $S_1$ pairs but any $S_2$ pairs. Then we can lower bound this term by

$$
\begin{aligned}
T_2 &= \sum_{t=1}^{T}\mathbb{1}\{\forall(i',k') \in S_1, (i',k') \notin m(t) \text{ and } \exists(i,k) \in S_2, (i,k) \in m(t)\} \\
&\geq \sum_{t=1}^{T}\sum_{i=N-\lfloor\frac{N}{2}\rfloor+1}^{N}\sum_{k=1}^{\lfloor\frac{N}{2}\rfloor}\mathbb{1}\{\forall(i',k') \in S_1, (i',k') \notin m(t) \text{ and } (i,k) \in m(t)\} \\
&= \sum_{i=N-\lfloor\frac{N}{2}\rfloor+1}^{N}\sum_{k=1}^{\lfloor\frac{N}{2}\rfloor}\sum_{t=1}^{T}\mathbb{1}\{\forall(i',k') \in S_1, (i',k') \notin m(t) \text{ and } (i,k) \in m(t)\} \\
&= \sum_{i=N-\lfloor\frac{N}{2}\rfloor+1}^{N}\sum_{k=1}^{\lfloor\frac{N}{2}\rfloor}\sum_{t=1}^{T}\left(\mathbb{1}\{(i,k) \in m(t)\} - \mathbb{1}\{(i,k) \in m(t), \exists(i',k') \in S_1, (i',k') \in m(t)\}\right) \\
&= \sum_{i=N-\lfloor\frac{N}{2}\rfloor+1}^{N}\sum_{k=1}^{\lfloor\frac{N}{2}\rfloor}\sum_{t=1}^{T}\mathbb{1}\{(i,k) \in m(t)\} \\
&\quad - \sum_{i=N-\lfloor\frac{N}{2}\rfloor+1}^{N}\sum_{k=1}^{\lfloor\frac{N}{2}\rfloor}\sum_{t=1}^{T}\mathbb{1}\{(i,k) \in m(t), \exists(i',k') \in S_1, (i',k') \in m(t)\} \\
&\geq \sum_{i=N-\lfloor\frac{N}{2}\rfloor+1}^{N}\sum_{k=1}^{\lfloor\frac{N}{2}\rfloor}\sum_{t=1}^{T}\mathbb{1}\{(i,k) \in m(t)\} - \left(\lfloor\frac{N}{2}\rfloor\right)\sum_{t=1}^{T}\mathbb{1}\{\exists(i',k') \in S_1, (i',k') \in m(t)\}
\end{aligned}
$$

$$= \sum_{i=N-\lfloor \frac{N}{2}\rfloor+1}^{N} \sum_{k=1}^{\lfloor \frac{N}{2}\rfloor} N_{i,k}(T) - \left(\lfloor \frac{N}{2}\rfloor\right) T_1 .$$

The first inequality is derived by Claim 1. The last inequality holds since at most $\lfloor \frac{N}{2}\rfloor$ player-arm pairs in $S_2$ can be simultaneously selected in one matching. And thus $T_1$ can be repeated count $\lfloor \frac{N}{2}\rfloor$ times, leading to

$$\sum_{i=N-\lfloor \frac{N}{2}\rfloor+1}^{N} \sum_{k=1}^{\lfloor \frac{N}{2}\rfloor} \sum_{t=1}^{T} \mathbb{1}\{(i,k) \in m(t), \exists (i',k') \in S_1, (i',k') \in m(t)\}$$

$$\leq \left(\lfloor \frac{N}{2}\rfloor\right) \sum_{t=1}^{T} \mathbb{1}\{\exists (i',k') \in S_1, (i',k') \in m(t)\} .$$

Therefore, we can lower bound the regret as

$$R(T,\nu,\pi) \geq (\frac{1}{2} + \Delta) \cdot T_1 + \Delta \cdot T_2$$

$$\geq (\frac{1}{2} - \lfloor \frac{N}{2}\rfloor \Delta)T_1 + \Delta \sum_{i=N-\lfloor \frac{N}{2}\rfloor+1}^{N} \sum_{k=1}^{\lfloor \frac{N}{2}\rfloor} \mathbb{E}\left[N_{i,k}(T)\right] \qquad (2)$$

$$\geq \Delta \sum_{i=N-\lfloor \frac{N}{2}\rfloor+1}^{N} \sum_{k=1}^{\lfloor \frac{N}{2}\rfloor} \mathbb{E}\left[N_{i,k}(T)\right] ,$$

where last inequality holds for $\Delta$ sufficiently small that $\Delta < 1/N$. This ends the proof.

### D.2 PROOF FOR LEMMA 2

$$\mathbb{P}\left(\mathcal{F}\right) = \mathbb{P}\left(\exists 1 \leq t \leq T, i \in [N], k \in [K] : |\hat{\mu}_{i,k}(t) - \mu_{i,k}| > \sqrt{\frac{6\log T}{N_{i,k}(t)}}\right)$$

$$\leq \sum_{t=1}^{T} \sum_{i\in[N]} \sum_{k\in[K]} \mathbb{P}\left(|\hat{\mu}_{i,k}(t) - \mu_{i,k}| > \sqrt{\frac{6\log T}{N_{i,k}(t)}}\right)$$

$$\leq \sum_{t=1}^{T} \sum_{i\in[N]} \sum_{k\in[K]} \sum_{s=1}^{t} \mathbb{P}\left(N_{i,k}(t) = s, |\hat{\mu}_{i,k}(t) - \mu_{i,k}| > \sqrt{\frac{6\log T}{s}}\right)$$

$$\leq \sum_{t=1}^{T} \sum_{i\in[N]} \sum_{k\in[K]} t \cdot 2\exp(-3\log T)$$

$$\leq 2NK/T ,$$

where the second last inequality is due to Lemma 3.

## E  DETAILS OF EXPERIMENTS

We give the reward matrix of the experiments in Section 5 here. As for another reward matrix with shape $10 \times 15$, all reward means are uniformly sampled from $[0,1]$ with no other constraints. All experiments are conducted on CPU.

$$U_{4\times 4} = \begin{bmatrix} 0.5 & 0.9 & 0.1 & 0.25 \\ 0.25 & 0.5 & 0.25 & 0.1 \\ 0.1 & 0.25 & 0.5 & 0.5 \\ 0.1 & 0.9 & 0.25 & 0.5 \end{bmatrix},$$

$$
U_{10\times10} =
\begin{bmatrix}
0.9 & 0.4 & 0.8 & 0.1 & 0.3 & 0.05 & 0.2 & 0.1 & 0.3 & 0.2 \\
0.4 & 0.3 & 0.3 & 0.1 & 0.2 & 0.3 & 0.4 & 0.4 & 0.3 & 0.4 \\
0.1 & 0.05 & 0.1 & 0.4 & 0.1 & 0.2 & 0.9 & 0.3 & 0.4 & 0.1 \\
0.05 & 0.1 & 0.9 & 0.2 & 0.9 & 0.75 & 0.1 & 0.9 & 0.25 & 0.05 \\
0.8 & 0.3 & 0.1 & 0.7 & 0.1 & 0.4 & 0.05 & 0.2 & 0.75 & 0.05 \\
0.4 & 0.05 & 0.3 & 0.7 & 0.05 & 0.1 & 0.25 & 0.75 & 0.6 & 0.05 \\
0.9 & 0.3 & 0.3 & 0.8 & 0.1 & 0.25 & 0.7 & 0.05 & 0.2 & 0.3 \\
0.3 & 0.1 & 0.4 & 0.25 & 0.05 & 0.9 & 0.25 & 0.1 & 0.05 & 0.4 \\
0.8 & 0.75 & 0.1 & 0.2 & 0.4 & 0.05 & 0.3 & 0.2 & 0.1 & 0.25 \\
0.4 & 0.4 & 0.9 & 0.7 & 0.25 & 0.2 & 0.05 & 0.1 & 0.4 & 0.25
\end{bmatrix}.
$$

# F DETAILS OF COMMUNICATION PHASE

In this section, we provide the details of communication phase in Algorithm 1 if we send information by bit. As the communication order can be decided as soon as the number of players is given, we focus on the communication between any player $i \in \mathcal{N}$ and player $j \in \mathcal{N}$, where $i < j$ in epoch $s$ in Algorithm 3. If player $i$ and $j$ collide in arm $i$, they receive digit 1 from each other; if they collide in arm $j$, they receive digit 0 from each other, otherwise, their current digit is different from one another. By such process, player $i$ can receive $\hat{\mu}_{j,k}(s)$ and player $j$ can receive $\hat{\mu}_{i,k}(s)$ for all $k$.

---

**Algorithm 3** Communication phase in Decentralized Fair Elimination (for players $i$ and $j$)

---

1: Players $i$ and $j$ transfer $\hat{\mu}_{i,k}(s)$ and $\hat{\mu}_{j,k}(s)$ for every $k \in \mathcal{K}$ into binary data, $\tilde{\mu}_{i,k}(s)$ and $\tilde{\mu}_{j,k}(s)$, respectively.
2: **for** $k = 1, 2, \ldots, K$ **do**
3:     **for** $\tau = 1, 2, \ldots, L$ **do**
4:         **if** the $\tau$'s digit of $\tilde{\mu}_{i,k}(s)$ is 1 **then**
5:             Player $i$ selects arm $i$.
6:         **else**
7:             Player $i$ selects arm $j$.
8:         **end if**
9:         **if** the $\tau$'s digit of $\tilde{\mu}_{j,k}(s)$ is 1 **then**
10:        Player $j$ selects arm $i$.
11:       **else**
12:          Player $j$ selects arm $j$.
13:       **end if**
14:    **end for**
15: **end for**

---

