# OpenReview forum: "Optimal Algorithm for Max-Min Fair Bandit"
_ICLR.cc/2025/Conference — Submitted to ICLR 2025_

### Official Review · Reviewer_ic7z · 2024-11-03

**Soundness:** 2
**Presentation:** 3
**Contribution:** 3
**Rating:** 6
**Confidence:** 3

**Summary:**

The paper presents a theoretical study for the Multi-Player Multi-Armed Bandit (MP-MAB) problem with a max-min fairness objective. In this scenario, multiple players choose from a set of arms, and collisions between players result in zero rewards. The goal is to maximize the reward for the player with the lowest reward. The authors propose a decentralized algorithm called Decentralized Fair Elimination (DFE), which improves the existing regret upper bound from $O(\log T \log \log T)$ to $O({\log T}/ {\Delta})$. Additionally, a matching lower bound is provided, demonstrating the optimality of the proposed algorithm. The effectiveness of the algorithm is also verified through numerical experiments.

**Strengths:**

1. The DFE algorithm addresses fairness without requiring a central coordinator, making it scalable to larger systems and addressing practical concerns in decentralized settings like wireless networks.

2. The algorithm proposes a novel exploration assignment strategy that ensures even exploration across player-arm pairs, which leads to efficient elimination of sub-optimal arms and reduces overall regret.

3. The authors provide both a theoretical analysis (including regret upper and lower bounds) and empirical validation through simulations.

**Weaknesses:**

1. In remark 1, the authors claim that "we could still use collisions to transmit information bit by bit, resulting in an additional constant length of information bits without the communication assumption." However, using collisions to transmit information, players should quantize UCB/LCB estimates to avoid potentially infinite communication length upon communication, as these numbers are often decimal numbers. To ensure that the elimination phase remains work despite the quantization errors, the required communication length would need to be on the order of $O(\log 1/\Delta)$. Given that the number of epochs is $O(\log T)$, this results in an additional term of $O((\log T \log 1/\Delta))$ in the regret bound.

2. Could the authors elaborate on why the inequality in line 698 holds? The paper does not provide any explanation for this, and additional details would be greatly appreciated. Additionally, I am unsure if the definition of $\Delta_{i,k} = \gamma^* - \mu_{i,k}$ is meaningful, as this value can be non-positive.

**Questions:**

See above.

---

> ### Author Response · Authors · 2024-11-20
> **Response to reviewer ic7z**
>
> 1. Additional term in communication cost.
>
> We thank the reviewer for pointing out this additional $\log 1/\Delta$ term in the communication cost. It is correct if we do not make the communication assumption. We will add this in the discussion about communication phase. The total communication cost is $N \log T \log (1 / \Delta)$, which still does not affect the leading term $O((N^2+K)\log T / \Delta)$.
>
> 2. Could the authors elaborate on why the inequality in line 98 holds?
>
> I guess you are asking the inequality in line 698, which is $2^s \leq 24 \log T / \Delta_{i,k}^2$. First, we analyze the pair $(i,k)$ which is not eliminated at epoch $s-1$, which means it has been selected at least $2^s$ number of times. Then conditioned on good event $\neg\mathcal{F}$, we have that
>
>  $\|\hat{\mu}\_{i,k} (s) - \mu\_{i,k} \| \leq \sqrt{\frac{6\log T}{2^s}} $.
>
>  Moreover, since the optimal player-arm pair (i’, k’) with max-min reward $\gamma^\ast$ is not eliminated, it is also selected at least $2^s$ number of times, we have that
>
>  $| \hat{\mu}\_{i’,k’}(s) - \mu\_{i’,k’} | \leq \sqrt{\frac{ 6 \log T}{2^s}}$.
>
> Since sub-optimal pair $(i, k)$ is not eliminated, we have that $2 \sqrt{\frac{6\log T}{2^s}} \geq \mu_{i’,k’} - \mu_{i,k} := \Delta_{i,k}$. Otherwise it must hold that $UCB_{i,k}(s) < LCB_{i’,k’}(s)$ and (i,k) will be eliminated after $s-1$ epoch. Rearranging the terms in the inequality in line 698.
>
> The definition of $\Delta_{i,k}$ is meaning for since we only analyze those sub-optimal pair (i, k) with $\mu_{i,k} < \gamma^\ast$, which guarantees the $\Delta_{i,k}$ is always positive. Recall that from the definition of $\gamma^\ast$ we know that the minimum reward in any matching can never greater than $\gamma^\ast$, thus we only care about the number of times of selecting those sub-optimal pairs $(i,k)$ with $\mu_{i,k} < \gamma^\ast$.

---

> > ### Comment · Reviewer_ic7z · 2024-11-26
> > **Re: Author Response**
> >
> > Thank you for your response. It effectively addresses all of my concerns, and I have adjusted my score accordingly. I recommend that the authors also incorporate the clarification regarding the inequality mentioned in line 698 into the revised paper.

---

### Official Review · Reviewer_fsnh · 2024-11-03

**Soundness:** 3
**Presentation:** 3
**Contribution:** 2
**Rating:** 5
**Confidence:** 3

**Summary:**

This paper addresses the multi-player multi-armed bandit (MP-MAB) problem, with the goal of finding a max-min fairness matching that maximizes the reward of the player receiving the lowest reward. The authors propose a decentralized fair elimination (DFE) algorithm to handle heterogeneous reward distributions and collisions between players. The algorithm improves the regret upper bound from $O(\log T \log \log T)$ to $O\left(\left(N^2+K\right) \log T / \Delta\right)$, where $\Delta$ is the minimum reward gap. Additionally, they provide a regret lower bound, demonstrating that the algorithm is optimal concerning key parameters.

**Strengths:**

1. The authors design a new phased elimination algorithm that improves max-min fairness by adaptively eliminating suboptimal arms and exploring the remaining ones. The algorithm achieves a regret upper bound of $\mathrm{O}\left(\left(\mathrm{N}^2+\mathrm{K}\right) \log \mathrm{T} / \Delta\right)$, which outperforms existing results.
2. The paper derives a tighter regret lower bound of $\Omega(\max (\{N^2, K\}) \log T / \Delta)$, which considers the parameters $N$, $K$, and $\Delta$, improving upon prior work.
3. Numerical experiments confirm the effectiveness of the DFE algorithm across different settings.

**Weaknesses:**

1. I appreciate the contribution of this work, which presents the first optimal result for the MP-MAB problem, aligning with the lower bound established here. However, compared to earlier studies, particularly those by Bistritz et al. (2020) and Leshem (2023), this study, which employs a classic elimination-based strategy, only improves the regret results by a factor of $\log\log T$. This improvement may not be particularly significant for smaller values of $T$. Does this work demonstrate advantages over others in terms of factors such as $N$ or $K$?

2. Recently, several studies have reported intriguing results on reducing communication costs in the MP-MAB problem for both competitive and cooperative settings. I believe the authors could strengthen this study by incorporating a communication-efficient algorithm. Can the authors provide a rigrous upper bound for the communication cost in this work and discuss the possibility to make it optimal? Did the work improve the communication cost compared to previous work?

**Questions:**

see weaknesses

---

> ### Author Response · Authors · 2024-11-20
> **Response to reviewer fsnh**
>
> We thank the reviewer for your valuable and detailed comments. Please see our response below.
>
> 1. Does this work demonstrate advantages over others in terms of factors such as $N$ or $K$?
>
> As the dependence of $N$ and $K$ in previous works, we note that the work of Leshem (2023) attains $O(N^3 \log T \log\log T)$ regret, where they assume $N=K$. And in the work of Bistritz et al. (2020), they can only get $O(\exp(N, K) \log T \log\log T)$ regret. Therefore our $O((N^2 + K) \log T / \Delta)$ regret not only gets the improvement over the term $T$, but also improves the dependence on term $N$ and $K$.
>
> Additionally, we highlight that the improvement over $T$ is not only reflected in removing the term $\log \log T$, but also removing a very large constant before the leading term, which could be exponentially large with $1 / \Delta, N, K$. This is because previous works  Bistritz et al. (2020); Leshem (2023) provide an explore-then-commit (ETC) method at each epoch $s$. Specifically, they let each player explores each arm $\log s$ times at the beginning of the epoch $s$, and then compute the max-min matching based on history observations in exploration phase. After that each player follows this matching in the following $2^s$ rounds. Their algorithms both only obtain an $O(\log T \log \log T)$ regret bound since they have to set an increasing length of exploration at each epoch. This design is to make sure the probability of computing a wrong max-min matching is bounded by $\exp(−s)$ when $s$ is sufficiently large that $\log s > 1/\Delta$, then the regret in the exploitation phase can be bounded. This design also raises the problem of a large constant to guarantee $\log s>1/\Delta$, which requires initial warm-up rounds is $O(\exp(1/\Delta))$, which could be very large when $\Delta$ is small enough. We handle this problem by applying the elimination method which eliminate sub-optimal player-arm pair efficiently. This assures that no forced explorations will happen in later epochs.
>
> 2, Can the authors provide a rigrous upper bound for the communication cost in this work and discuss the possibility to make it optimal? Did the work improve the communication cost compared to previous work?
>
> Thanks for pointing out the communication cost in our algorithm. Here we give a rigrous analysis for it. If the minimum reward gap between max-min value $\gamma^\ast$ is $\Delta$, then the length of each communication phase is bounded by $N \log (1/\Delta)$. Here $\log (1 / \Delta)$ is the length of transmitting a reward’s information by bit and through collisions. We only need the bit length with $\log (1 / \Delta)$ since it is enough to distinguish two pairs with gap larger than $\Delta$.
>
> Then the total communication cost is $N \log T \log (1 / \Delta)$. We also note that the communication cost in Leshem (2023) is $\frac{3}{2} N^3 \log (1/\Delta) \log T$, and the communication cost in Bistritz et al. (2020) is $\exp{N, K}$. Additionally, we note that $\Delta$ in their works is the minimum reward gap among all player-arm pairs, whereas in our work $\Delta$ is only the minium reward gap between $\gamma^\ast$. Thus we also significantly improve the communication cost compared with previous works.
>
>  We believe that if the algorithm has to convey information of reward’s estimation, then our algorithm is optimal since we make players communication with each other by bit and through collisions, which is the most efficient way to communication as far as we know. We leave it as an interesting future work to design an algorithm with minimum communication cost.

---

> > ### Comment · Reviewer_fsnh · 2024-11-26
> >
> > Thanks for the detailed explanation. I have raised my score.

---

> > > ### Author Response · Authors · 2024-12-01
> > >
> > > Thank you for your feedback and for updating us on the score. We genuinely appreciate your insightful and constructive comments. Please let us know if you have any further concerns or questions, we will be delighted to clarify them.

---

### Official Review · Reviewer_RJ7b · 2024-11-04

**Soundness:** 3
**Presentation:** 2
**Contribution:** 3
**Rating:** 6
**Confidence:** 3

**Summary:**

This paper considers max-min fair bandit, an important variant of multi-player multi-armed bandit problem where fairness means to maximize the reward of the player who receives the lowest reward. Existing work in max-min fair bandit suffer from large regret and heavy assumptions. The authors give tight regret bound for the bandit problem that is optimal with respect to all parameters. Special case for the lower bound is provided. The work closes the gap of the man-min fair bandit problem.

**Strengths:**

This paper fills an important gap in existing max-min bandit literature. Tight regret bound is proved and special case is provided to demonstrate the lower bound. The improvement is significant compared to existing work. The regret bound is tight in all parameters. The algorithm is splitted into three phases with detailed algorithmic and graphical illustrations.

**Weaknesses:**

The main section of the decentralized fair elimination algorithm is a bit hard to read. It would be good if the authors can highlight the novelty part of the algorithm and clearly demonstrate how the three steps of the main algorithm contributes to the regret and which dominates the regret. The elimination phase is a commonly used strategy. The exploration phase is new, but it does not seem to directly contribute to the improvement of overall regret. The communication phase is also seen in the matching bandit literature.

**Questions:**

• Line 90 What is the doubling trick?
• Line 242 about the second type of elimination – unclear. How to determine the assertion of “with high probability”?
• If there is no elimination phase, how the regret is affected? Line 310 is confusing.
• Can you give intuitive idea / sketch proof to Thm 1, which is the key result for the whole paper, to explain how the three steps contribute to the regret bound?

-------------------------After Rebuttal-----------------------------
Thank you for the response. This helps me better understand the technical contribution of the paper. I see it's a theoretically interesting paper. I increased my confidence in my assessment.

---

> ### Author Response · Authors · 2024-11-20
> **Response to reviewer Rj7b**
>
> We thank the reviewer for your valuable and detailed comments. Please see our response below.
>
> 1. Highlight the novelty part of the algorithm and clearly demonstrate how the three steps of the main algorithm contributes to the regret and which dominates the regret. Give intuitive idea / sketch proof to Thm 1, which is the key result for the whole paper, to explain how the three steps contribute to the regret bound?
>
> We highlight that the novelty part of our proposed algorithm is the exploration phase with carefully designed exploring matching set given those non-eliminated player-arm pairs. More specifically, we improve the total exploration times in one cycle (explore all non-eliminated player-arm pairs) from a naive bound $NK$ to the optimal bound $N^2 + K$. This is also the key to match the lower bound. Elimination phase guarantees no sub-optimal player-arm pair will be selected too many times, and the optimal design of exploration set guarantees the difference in the number of times pairs are selected in the exploration set will not be too large. These two designs lead to the final optimal regret bound.
>
> 2. Line 90 What is the doubling trick?
>
> ‘’Doubling trick’’ means the length of exploration phase increases doubles each time. By this design we can control the total communication times by $O(\log T)$, and ensure that the number of additional explorations does not exceed twice the necessary number of explorations.
>
> 3.Line 242 about the second type of elimination – unclear. How to determine the assertion of “with high probability”?
>
> The second type of elimination means (j, k) will not occur in the optimal matching set if (j, k) does not exist in any matching set with UCB greater than $\underline{\gamma}_s$, and thus it will be eliminated. Here “with high probability” means if UCB is greater than $\underline{\gamma}_s$, then with high probability the minimum reward of the given matching is smaller than the optimal max-min reward.
>
> 4. If there is no elimination phase, how the regret is affected? Line 310 is confusing.
>
> If there is no elimination phase, all player-arm pairs will be selected the same number of times. Denote the minimum reward gap between the pair and max-min reward $\gamma^\ast$ as $\Delta$, and the gap between $\gamma^\ast$ and a sub-optimal pair $(i, k)$ is $\Delta_{i, k}$. Then the number of times of selecting $(i, k)$ is $O(\log T / \Delta^2)$ without elimination phase, and the regret caused by selecting $(i, k)$ is $\Delta_{i, k} O(\log T / \Delta^2)$, which can only be bounded by $ O(\log T / \Delta^2)$ since $\Delta < \Delta_{i, k}$. However, if we utilize the elimination phase, the number of times of selecting $(i, k)$ can be bounded by $O(\log T / \Delta_{i, k}^2)$, and thus the regret caused by selecting $(i, k)$ is $O(\log T / \Delta_{i, k})$. In general, by elimination phase, we can improve the regret by $O(1 / \Delta)$.

---

### Official Review · Reviewer_FPek · 2024-11-13

**Soundness:** 3
**Presentation:** 2
**Contribution:** 3
**Rating:** 6
**Confidence:** 3

**Summary:**

This work studies a multi-player multi-armed bandit problem with heterogeneous reward and collision.
This paper aims to find a fair bandit algorithm that matches each player to a distinct arm while maximizing the reward of the player who receives the smallest reward.
This paper provides a max-min regret lower bound of $\Omega(\max(N^2, K) \log T/\Delta)$.
The authors propose the decentralized fair elimination (DFE) algorithm that guarantees the exploration of all valid player-arm pairs by constructing matchings, controls the communication times by the doubling trick, and eliminates player-arm pairs whose upper confidence bounds are smaller than the lower confidence bound of the current max-min value.
The authors show that DFE achieves $O((N^2+K)\log T/\Delta)$ max-min regret.
There is also an empirical study of the performance of the proposed algorithm compared to prior decentralized competitive MP-MAB algorithms.

**Strengths:**

- The problem studied (fairness in MP-MAB) is important and interesting.
- The algorithmic designs of exploration and elimination procedures are interesting.
- This paper conducts both theoretical and empirical studies.
- I appreciate the diagrams for algorithmic design illustration.

**Weaknesses:**

- The claim that the proposed algorithm is optimal concerns me because the upper bound $O((N^2+K)\log T/\Delta)$ does not match exactly with the lower bound $\Omega(\max(N^2, K) \log T/\Delta)$ in terms of $N$ and $K$. The proposed algorithm is definitely near-optimal, but I am concerned about claiming it as optimal.
- The readability of this paper can be further improved. Notations can be introduced more clearly. For example:
    - A matching $m$ is first introduced as "matching set" $m(t)$, but later more used as matching $m$ or $m_i$
    - The definition of regret in this paper is very different from that in multi-armed bandit literature. I would suggest the authors make this point clearer in the paper.
    - The last sentence on the first page is too long.
    - $\mathcal{P}$ has always denote player-arm set in the rest of the paper. However, it is used differently in Section 4.
    - Claim 1 only holds for the instance described in Section 4. This point should be made clear, as in Lemma 1.
    - Figure 5 is not very color-blind friendly or black-white printable.

**Questions:**

- Is the $O(\log T \log \log T)$ regret bound of Bistritz et al. (2020) analyzed against the max-min regret same as defined in Section 2? If not, it seems unfair to compare with it in the introduction.
- Is there any hyperparameter for the proposed algorithm or the benchmarks the authors set in the experiments? If so, please add them to the paper to increase replicability.

---

> ### Author Response · Authors · 2024-11-20
> **Response to reviewer FPek**
>
> We thank the reviewer for your valuable and detailed comments. Please see our response below.
> 1. Does the upper bound match the lower bound?
>
> In this paper, we propose the algorithm attaining the upper bound $O((N^2+K) \log T / \Delta)$ and provide the analysis with lower bound $\Omega(\max(N^2,K) \log T / \Delta)$. Here we confirm that these two bounds are exactly in the same order, since we can see that $\max(N^2, K) \log T / \Delta \leq (N^2+K) \log T / \Delta \leq 2\max(N^2, K) \log T / \Delta $. This shows that the upper bound exactly matches the lower bound with respect to term $N, K, T, \Delta$. Therefore, we indeed design an optimal algorithm for this problem.
>
> 2. Is the $O(\log T \log\log T)$ regret bound of Bistritz et al. (2020) analyzed against the max-min regret the same as defined in Section 2?
>
> We study the exact same setting as in the work of Bistritz et al. (2020). Moreover, we do not assume each player must have different rewards over arms like Bistritz et al. (2020). Thus we analyze a more general setting and the definition of regret is the same as those compared works.
>
> 3. Is there any hyperparameter for the proposed algorithm or the benchmarks the authors set in the experiments?
>
> Indeed our algorithm is parameter-free, thus it is easy to follow our algorithm’s description to reproduce the same performance. The hyperparameters of the benchmarks are the same as they stated in their paper, and we will restate them in the updated version.
>
> 4. Thanks for pointing out these unclear notations, we will fix them in the updated version.

---

> > ### Comment · Reviewer_FPek · 2024-11-26
> >
> > The authors' response addressed my concern. I have raised the rating. I would encourage the authors to add the above discussion to the paper to enhance clarity.

---

### Author Response · Authors · 2024-11-25
**Official Comment by Authors**

Dear Reviewers:

We thank you once again for your careful reading of our paper and your constructive comments and suggestions. We would appreciate it if you could let us know whether all your concerns are addressed. We are also happy to answer any further questions in the remaining discussion period.

Best, Authors.

---

### Author Response · Authors · 2024-11-28

Dear Reviewers:

We appreciate the your insightful thoughts, and we will definitely integrate these comments and discussions into our next version. Thanks!

Best, Authors

---

### Meta-Review · Area_Chair_uMo6 · 2024-12-20

**Metareview:**

This paper looks at a variant of the multi-player multi-armed bandit setting where the objective is to minimize some min-max regret instead of the usual cumulative one.

The reviewers and I are not that thrilled by the results and the algorithm, as they are not straightforward, but relatively similar than the existing ones in the literature.
It is true that this paper improves a log(T)loglog(T)  bounds into a log(T) regret bound, but this does not necessarily says that achieving it was difficult (rather than questioning the quality of the first paper).

All in all, I do not think this paper reaches the ICLR bar, even though it is, as far as I can see, correct and midly interesting.

**Additional Comments On Reviewer Discussion:**

All reviewers were lukewarm, and none of them decided to champion this paper for acceptance. As I was not thrilled either, the conclusion was clear

---

### Decision · Program_Chairs · 2025-01-22

Reject